# The implications of maintaining Earth's hemispheric albedo symmetry for shortwave radiative feedbacks

Aiden R. Jönsson[1,2] and Frida A.-M. Bender[1,2]

[1]Department of Meteorology, Stockholm University, 106 91 Stockholm, Sweden
[2]Bolin Centre for Climate Research, Stockholm University, 106 91 Stockholm, Sweden

**Correspondence:** Aiden R. Jönsson (aiden.jonsson@misu.su.se)

**Abstract.** The Earth's albedo is observed to be symmetric between the hemispheres on the annual mean timescale, despite the clear-sky albedo being asymmetrically higher in the northern hemisphere due to more land area and aerosol sources; this is because the mean cloud distribution currently compensates for the clear-sky asymmetry almost exactly. We investigate the evolution of the hemispheric difference in albedo in CMIP6 coupled model simulations following an abrupt quadrupling of $CO_2$ concentrations, to which all models respond with an initial decrease of albedo in the northern hemisphere (NH) due to loss of Arctic sea ice. Models disagree over whether the net effect of NH cloud responses is to reduce or amplify initial NH albedo reductions. After the initial response, the evolution of the hemispheric albedo difference diverges among models, with some models remaining stably at their new hemispheric albedo difference, and others returning towards their pre-industrial difference primarily through a reduction in SH cloud cover. Whereas local increases in cloud cover contribute to negative shortwave cloud feedback, the cross-hemispheric communicating mechanism found primarily responsible for restoring hemispheric symmetry in the models studied, implies positive shortwave cloud feedback.

## 1   Introduction

The Earth's albedo is hemispherically symmetric to a high degree; the northern hemisphere (NH) minus southern hemisphere (SH) difference in annual mean hemispheric albedo (henceforth referred to as asymmetry) has been on the order of 0.1 W m$^{-2}$ for the past two decades (Datseris and Stevens, 2021; Jönsson and Bender, 2022). This was first noted, although with greater uncertainty, during the first generation of satellite observations of Earth's radiative energy balance (Vonder Haar and Suomi, 1971), and persists without detectable trends in modern satellite observations (Stevens and Schwartz, 2012; Voigt et al., 2013; Stephens et al., 2015; Datseris and Stevens, 2021; Jönsson and Bender, 2022). This is possible because in the annual mean all-sky albedo, clouds compensate for the clear-sky albedo asymmetry that exists due to differences in surface properties and aerosol loading between the NH and SH (Stephens et al., 2015; Jönsson and Bender, 2022; Diamond et al., 2022). Climate models have a large spread of albedo asymmetry (Stephens et al., 2015; Jönsson and Bender, 2022), but the variability of asymmetry in model simulations is low, with most coupled models exhibiting relatively small changes between pre-industrial (PI) and present-day (PD) historical asymmetries (Jönsson and Bender, 2022).

The clear-sky hemispheric albedo asymmetry is determined mostly by contributions from the clear-sky atmosphere (Stephens et al., 2015; Jönsson and Bender, 2022), pointing to a strong influence from aerosols in the NH in leading to a presently higher NH than SH clear-sky albedo (Diamond et al., 2022). In higher latitudes, the surface contributes a greater share of the albedo (Stephens et al., 2015). Because of aerosol emission drawdown and changes in the cryosphere due to global warming, the clear-sky hemispheric albedo asymmetry is likely to change in the near future (Diamond et al., 2022).

Higher cloud amount in the SH subtropics as well as higher cloud amount and cloud albedo in the SH midlatitudes than in their NH counterparts compensate for both the clear-sky albedo asymmetry and higher cloud amount in the NH tropics than in the SH tropics (Bender et al., 2017). Although it has been shown that the tropical maximum in deep convective cloud cover following the position of the inter-tropical convergence zone (ITCZ) could offer some compensation to a hemispheric difference in albedo by shifting into the darker, warmer hemisphere (Voigt et al., 2013, 2014), tropical clouds have been understood not to have a major role in determining the hemispheric albedo symmetry, since the tropical maximum in cloud cover is located in the NH. Thus, extratropical cloud cover – particularly in the SH midlatitudes – has been highlighted as important for maintaining the hemispheric albedo symmetry in the annual mean and beyond (Datseris and Stevens, 2021; Jönsson and Bender, 2022; Rugenstein and Hakuba, 2021), while variability in tropical cloud cover has been found to contribute to variability in the albedo asymmetry time series (Jönsson and Bender, 2022). Changes in SH extratropical clouds in response to anthropogenic forcing would inevitably impact the hemispheric albedo symmetry; furthermore, clouds in the SH extratropics are responsible for a large share of the positive shift in model estimates of SW cloud feedbacks from CMIP5 to CMIP6 (Zelinka et al., 2020). However, constraining the magnitude of this shift and the representation of SH extratropical clouds in models is made challenging due to a lack of observations (Ceppi and Hartmann, 2015; Gettelman et al., 2020).

The Earth's albedo is to a large degree determined by contributions from clouds, accounting for over half of the upwelling shortwave (SW) radiative fluxes at the top of the atmosphere (TOA) in the global mean. This means that the planetary albedo is relatively sensitive to changes in cloud properties and coverage with a changing climate. The sum effect of clouds on changes in planetary albedo and thus reflected SW radiation on Earth's radiative balance at the TOA in response to change in temperature is referred to as the total SW cloud radiative feedback, and its spatial distribution as well as its global mean magnitude is the greatest source of uncertainty in estimating the climate's sensitivity to $CO_2$ forcing (Forster et al., 2021). The spread of SW cloud radiative feedbacks estimated by coupled models has increased in the latest phase of Coupled Model Intercomparison Project (CMIP6) compared to the previous (CMIP5), and its average value has increased from slightly negative in CMIP5 to slightly positive in CMIP6 (Zelinka et al., 2020). Observational constraints also support this positive SW cloud radiative feedback estimate (Ceppi and Nowack, 2021; Forster et al., 2021). The hemispheric albedo symmetry is thereby relevant in addressing a significant source of uncertainty in constraining estimates of climate sensitivity: understanding any mechanisms that might maintain this symmetry can aid in estimating the magnitude and distribution of the total SW cloud radiative feedback.

While there is no known physical mechanism or explanation for the observed hemispheric albedo symmetry, it is important to pose the question: what would a mechanism for maintaining a hemispheric albedo symmetry entail for climate? Given that there is no observed trend in the hemispheric difference in albedo despite changes in the global radiative energy balance and despite global changes in albedo (Stephens et al., 2022), the hemispheric symmetry is at least robust throughout the satellite

record. In this study, we investigate the implications for Earth's climate if its albedo were forced out of its current hemispheric symmetry due to warming processes, to guide an exploration of possibilities for changes in the global cloud distribution in a changing climate.

We examine possible pathways for the Earth's albedo symmetry response to warming using climate models, and discuss how these pathways for hemispheric albedo differences in a perturbed climate manifest in terms of changes to the cloud distribution, heat transport, energy balance, and warming. To this end, we use simulations from an ensemble of coupled atmosphere-ocean and earth system models from the Coupled Model Intercomparison Project phase 6 (CMIP6) (Eyring et al., 2016) in which $CO_2$ concentrations are abruptly quadrupled from PI levels. These idealized single-forcing experiments allow for study of the evolution of albedo asymmetry in models in response only to greenhouse gas (GHG)-forced warming, without consideration of aerosol forcing that is presently significant but can be expected to be much smaller than the $CO_2$ forcing in the future if ongoing aerosol emission drawdown continues (Myhre et al., 2015; Szopa et al., 2021). We show how modeled albedo asymmetries evolve as the climate warms, and categorize model behavior by symmetry-maintaining responses (Section 3.1). We then characterize potential albedo symmetry-maintaining mechanisms and how strongly they act among the models (Sections 3.2 and 3.3). Finally, we describe the implications that symmetry-maintaining mechanisms have for the strength of SW cloud radiative feedbacks (Section 3.4), and discuss the realism of these mechanisms (Section 4).

## 2 Materials and methods

### 2.1 Model output

In this analysis, we use CMIP6 abrupt, strong forcing (*abrupt-4xCO2*) experiments as well as simulations of PI conditions (*piControl*) and those using best estimates of past forcings (*historical*) (Eyring et al., 2016). Abrupt forcing simulations can be used to estimate a model's equilibrium climate sensitivity (ECS) by regressing its global mean temperature response against its net TOA radiation imbalance; this slope yields the effective climate sensitivity (EffCS), a first-order estimate of ECS (Gregory et al., 2004). This method requires a reasonable amount of simulation time (the minimum for CMIP6 participation being 150 years) compared to the millennia that are needed for a model's climate to equilibrate and yield a calculation of the ECS (Rugenstein et al., 2020), and thus includes a high number of models. The abrupt forcing simulations allow sequences of events in the adjustment of the climate system to be studied as they occur on different timescales. In this study, we consider one realization (*r1i1p1f1*) each from 34 models (listed in Table 1) and discuss their evolution over 150 years of simulation time following the onset of forcing. We use one realization (*r1i1p1f1*) of historical simulations for each model to estimate values of PD conditions over the years 2000-2014, to compare to Clouds and the Earth's Radiant Energy System, Energy Balanced and Filled (CERES EBAF) radiative fluxes and Moderate Resolution Imaging Spectroradiometer (MODIS) cloud fraction obtained from observations over March 2000-February 2015 (Loeb et al., 2018). We also make use of estimates of model radiative feedback strengths published by Zelinka et al. (2022) and model ECS published by Meehl et al. (2020). We choose only models where all radiative flux variables were available. Certain models are excluded when output needed for a given comparison was not available; variable output coverage is specified in Appendix A.

## 2.2 Data processing

We focus our analysis on modeled reflected SW radiative fluxes at the TOA ($F_{TOA}^{\uparrow}$) and albedo $\alpha$ in all- and clear-sky conditions, as well as the SW cloud radiative effect (CRE), defined as the difference between clear- and all-sky $F_{TOA}^{\uparrow}$:

$$\text{SW CRE} = F_{TOA,\text{ clear}}^{\uparrow} - F_{TOA,\text{ all}}^{\uparrow}, \tag{1}$$

so that a negative CRE implies TOA cooling. NH minus SH hemispheric differences in $F_{TOA}^{\uparrow}$ are referred to as *asymmetry*, and hemispheric differences in other values are denoted with $\delta_{HD}$. Differences in time are denoted with $\Delta$. Area averages are calculated using meridional weights given by the cosine of latitude, i.e. assuming a spherical Earth model. In calculating time averages, we weight CERES EBAF time averages by the length of months in days, but we do not weight monthly averages in models by the length of the month due to differences in model calendars; we motivate this with the assumption that differences in time averages among the 34 models presented here should arise primarily from differences in the models themselves and secondarily by the 5-day ($\sim$1%) spread in model calendar years. Where correlation statistics are given, the correlation is significant at the 99% confidence level ($p < 0.01$) unless otherwise stated.

To estimate meridional heat transport (MHT) and its components, we use monthly mean TOA and surface energy fluxes following the methods outlined in Donohoe et al. (2020). We show only the tendency of meridional redistributions of energy absorbed by the climate system, as the modeled climate systems in these simulations are not in equilibrium. The implied total (ocean plus atmosphere) MHT is assumed to be driven by the meridional distribution of TOA energy imbalance, and the implied atmospheric heat transport (AHT) is assumed to be driven by the energy input into the atmosphere, or difference between the TOA energy input and the total surface energy input.

## 3 Results

### 3.1 Modeled albedo asymmetry responses to CO$_2$ forcing

Figure 1 depicts the time evolution of modeled asymmetries relative to their PI mean hemispheric albedo difference. In all models, asymmetry immediately becomes more negative following a reduction of clear-sky albedo asymmetry, which occurs primarily in the first 50 years after the forcing is applied; from here on, we will refer to the period between 30-50 years into the experiment as 'Mid', and use its mean conditions as representative of the model state after the initial albedo asymmetry response. After the 'Mid' period, the evolution of the asymmetry time series diverges among the models, with some models' asymmetry remaining relatively stable, and others recovering towards their PI mean asymmetry; this divergence is driven by cloud responses. We will henceforth refer to the years 130-150 as the 'End' period. The divergence of asymmetry responses after 'Mid' can be seen in the distribution of asymmetry differences between 'End' and 'Mid' (Figure 1c). The correlation between clear- and all-sky asymmetry responses between 'Mid' and PI conditions is strong ($R^2 = 0.78$) and weak between the 'End' and 'Mid' periods ($R^2 = 0.33$), illustrating the role of clouds in the divergence of asymmetry responses following the initial response. In two models, EC-Earth3-Veg and EC-Earth3-AerChem, asymmetry continues to strengthen in the negative

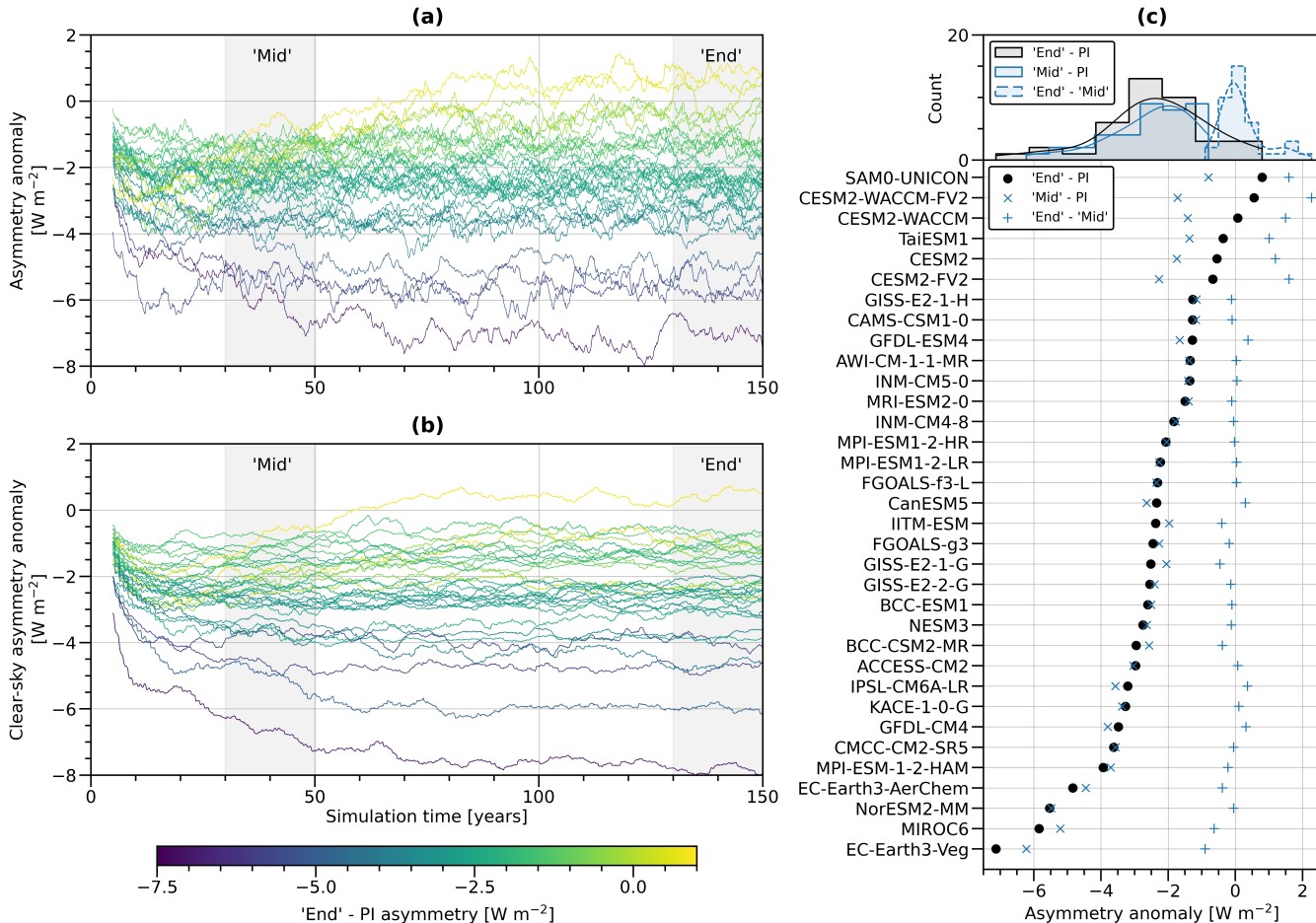

**Figure 1.** Left: time series of 5-year running mean modeled (a) all-sky and (b) clear-sky asymmetry responses in abrupt-4xCO2 simulations. The color scale of time series lines is representative of the total 'End' minus PI albedo asymmetry difference. Right: (c) differences in mean asymmetries between periods, listed in order of 'End' minus PI asymmetry changes. Histograms (with 8 bins; also shown as curves smoothed with a Gaussian kernel) of the distributions of asymmetry responses by period are shown at the top.

direction due to continued NH darkening. By 150 years, several models have recovered their PI mean asymmetry, with some overcompensating towards a more positive asymmetry than in PI conditions.

By 'End', modeled responses in hemispheric differences in net TOA energy inputs are also anticorrelated with asymmetry responses ($R^2 = 0.79$, 0.41 and 0.39 for all-sky, clear-sky and CRE asymmetry responses, respectively), with a change of -0.92 W m$^{-2}$ for each W m$^{-2}$ difference in the all-sky asymmetry response. A positive disturbance in the NH-SH hemispheric mean difference in net TOA energy input would induce anomalous southward cross-equatorial heat transport — or, if cross-equatorial transport is not changed, hemispherically asymmetric warming and/or deep-ocean heat storage.

Figure 2 includes profiles of zonal mean changes in all- and clear-sky reflected radiation as well as SW CRE throughout the simulations; composite maps and maps of inter-model spread for these variables are shown in Supplementary Figures S1-S3. Figure 2 shows that the initial negative asymmetry response is primarily due to a reduction in NH high-latitude clear-sky albedo (Figure 2a) and secondarily to changes in NH cloud cover (Figure 2b). Polar albedo reductions in the SH are generally smaller than those in the NH. High-latitude NH albedo losses are consistent with other studies on the amplified warming of the Arctic in response to GHG forcing (e.g. Hahn et al. 2021; Sledd and L'Ecuyer 2021), and can be ascribed to sea ice loss. While models agree on the direction of SW CRE changes in the Arctic, they disagree on the direction of SW CRE changes at other latitudes in the NH (Figure 2c, f); the NH albedo responses will be discussed in Section 3.2. Models that recover towards their PI mean asymmetry by the end of the simulation do so primarily by weakening negative SH midlatitude SW CRE beyond 'Mid' (Figure 2i); these responses will be discussed in Section 3.3. All models generally see a weakening in subtropical SW CRE in both hemispheres throughout the 150 years.

It is important to note that we present the evolution of modeled albedo asymmetry relative to PI conditions to study its potential response to warming, and there is a large spread in PI mean hemispheric albedo differences among models (Figure 1c) (Jönsson and Bender, 2022; Diamond et al., 2022; Rugenstein and Hakuba, 2021). However, we do not see any consistent or robust dependence of the asymmetry response upon PI mean asymmetries across models, which is in agreement with Rugenstein and Hakuba (2021).

## 3.2  Initial NH cloud compensations to clear-sky darkening

Although the asymmetry response following the initial NH surface darkening is unanimous among models in terms of sign, there is spread in the magnitude of this asymmetry response. Models disagree on whether cloud responses strengthen or dampen the NH clear-sky albedo reduction seen in the asymmetry time series. Disturbances in the hemispheric albedo symmetry with warming may be reduced by clouds when they buffer clear-sky albedo reductions; we will refer to these compensations as *local compensations*, and present modeled NH cloud responses in order to interpret the degree of local compensations in the hemisphere where clear-sky albedo reductions are greatest. To understand where clouds and clear-sky albedo changes are impacting the initial NH darkening the most, changes in the clear- and all-sky reflected SW radiation, as well as SW CRE, in three ranges of latitudes are presented in Figure 3. Figure 2 shows that most of these responses in the NH occur primarily by the end of the 'Mid' period and evolve less thereafter; changes after 'Mid' occur mostly in high latitudes.

Poleward of 60° N, models agree that clouds reduce the impact of the loss of Arctic ice cover on Arctic planetary albedo, as has been detailed in Sledd and L'Ecuyer (2021), although they disagree on the magnitude of this cloud compensation for surface albedo darkening. However, the overall impact of clouds on NH albedo varies between models, despite masking the surface albedo reduction in higher latitudes. The response in Arctic SW CRE has only weak bearing on the overall change in NH SW CRE ($R^2 = 0.15$ with $p = 0.02$) and virtually none on the total NH albedo change ($R^2 < 0.01$ with $p = 0.95$) between PI conditions and 'Mid'.

By 'Mid', SW CRE is weakened among all models to varying degrees in the NH subtropics to midlatitudes, leading to a reduction in albedo. However, models disagree over whether clouds in the NH tropics amplify or compensate for the reduction

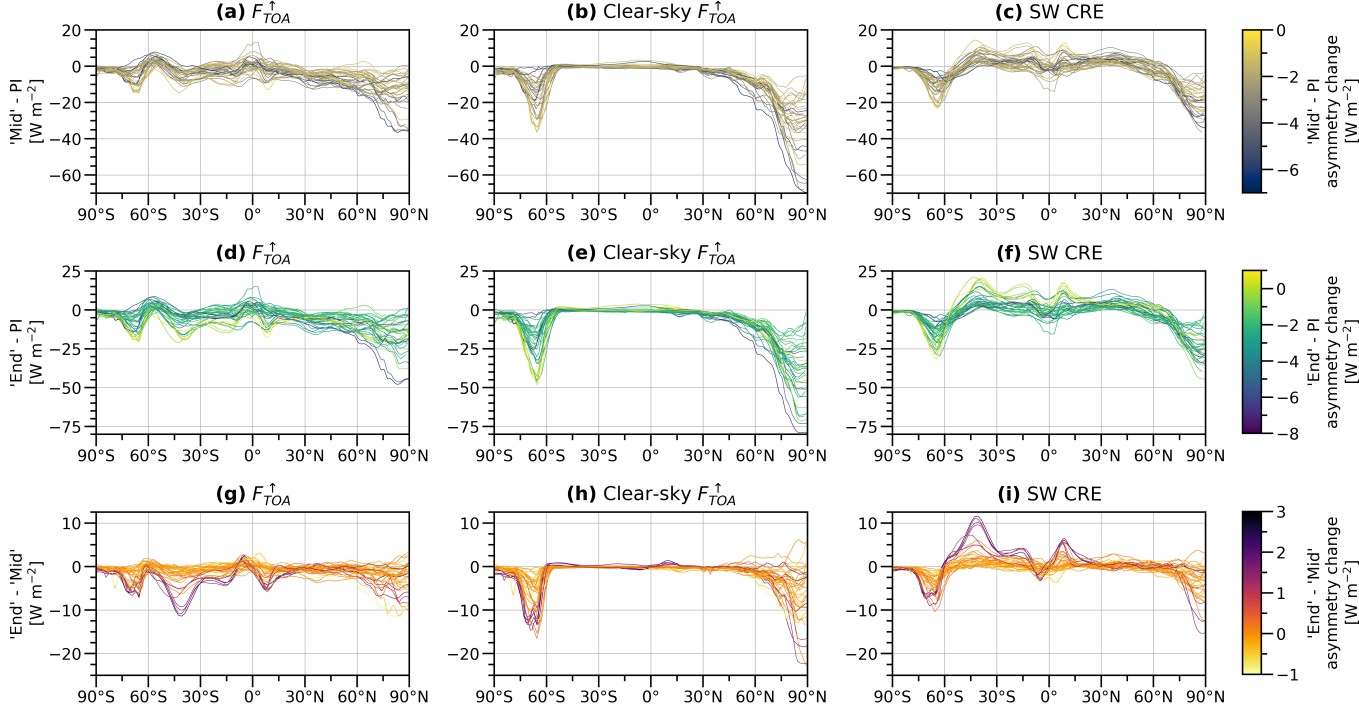

**Figure 2.** Meridional profiles of differences in zonal mean all- and clear-sky $F_{TOA}^{\uparrow}$, as well as SW CRE between (a-c) 'Mid' and PI conditions, (d-f) 'End' and PI conditions, and (g-i) 'End' and 'Mid'. Color scales represent the change in modeled mean hemispheric asymmetry between those periods.

in albedo. In some models, changes in tropical cloud cover cause SW CRE to strengthen and increase albedo; these models have a weaker overall reduction in NH albedo and thus weaker initial albedo asymmetry response.

The sum effect of these changes is that models disagree on whether clouds would amplify or reduce the initial NH albedo decrease caused by changes in clear-sky albedo during warming, depending primarily on the direction and strength of subpolar SW CRE changes, and secondarily on the degree of strengthening negative SW CRE in the Arctic. Furthermore, even models with little change in subtropical and midlatitude SW CRE and compensating tropical SW CRE do not compensate fully for NH clear-sky albedo reductions between PI conditions and 'Mid'.

### 3.3 Subsequent SH cloud responses to warming

One way to maintain a hemispheric albedo symmetry after perturbing the clear-sky hemispheric albedo difference would be that, when one hemisphere darkens, the other darkens as well. Models disagree on SH albedo reductions, which occur primarily in the SH extratropics, causing the divergence in modeled hemispheric albedo asymmetry after 'Mid' noted in Section 3.1 and leading to some models recovering towards their PI mean albedo asymmetry; we will refer to compensations to the NH

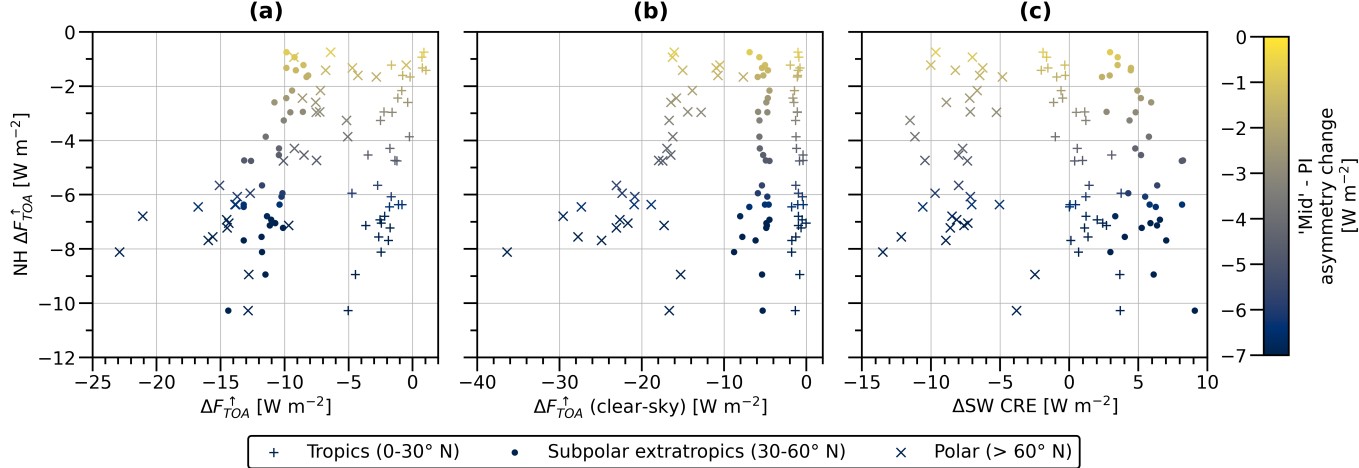

**Figure 3.** Area mean differences in (a) all-sky and (b) clear-sky $F_{TOA}^{\uparrow}$, as well as (c) SW CRE, in three ranges of latitudes plotted against the NH mean change in FTOA between 'Mid' and PI conditions. The color scale is representative of the 'Mid' minus PI albedo asymmetry difference.

darkening offered by SH albedo reductions as *remote compensations*. In this section, we use modeled change in asymmetry between 'End' and 'Mid' as a measure of each model's remote compensation to the initial asymmetry response.

Comparing changes in SH midlatitude SW CRE against cloud properties (see supplementary Figure S4) reveals that reductions in cloud fraction primarily drive weakening SW CRE ($R^2 = 0.95$), which out-compete increases in in-cloud cloud
water content and changes in phase partitioning (greater fraction of liquid water content) that would otherwise make SW CRE more negative (Mülmenstädt et al., 2021) in some models. The reductions in SH midlatitude cloud cover reported here are also discussed in Gjermundsen et al. (2021), who investigated the differences in warming responses between two related models, CESM2 and NorESM2. These differences are attributed to the different ocean component models, in which Southern Ocean (SO) deep convection slowing occurs more quickly in CESM2 than in NorESM2, allowing sea surface temperatures (SSTs) to
increase more quickly and thus impact low cloud cover.

Beyond a divergence in SH extratropical SW CRE among models, we find a relationship between the evolution of climate and albedo at high latitudes in the SH with changes in SH extratropical cloud cover between 'Mid' and 'End'. Figure 4 shows changes in relevant SH polar climate variables plotted against changes in asymmetry between 'Mid' and 'End'. As the atmosphere warms (Figure 4a), water vapor content increases in the SH (Figure 4b), leading to greater poleward moisture
transport and greater precipitation in the high SH latitudes. In the midlatitudes and sea ice zone ($\sim$50-75° S), this manifests increasingly as liquid precipitation while snowfall is reduced; on the Antarctic continent, this mostly manifests as increased snowfall. SW CRE becomes more strongly negative over the Antarctic sea ice zone, which is to be expected as the highly reflective ice-covered surface contributions to albedo are reduced during warming.

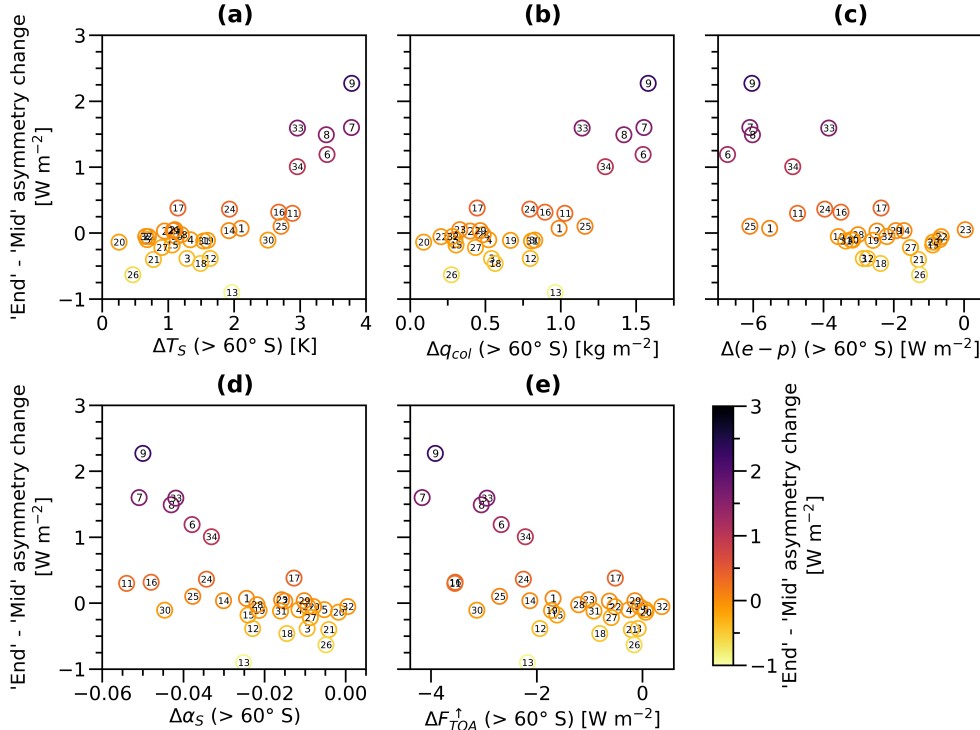

**Figure 4.** Mean asymmetry changes plotted against SH polar (> 60° S) area mean changes in (a) near-surface air temperature $T_S$, (b) vertically integrated atmospheric moisture content $q_{col}$, (c) evapotranspiration minus precipitation $(e-p)$ expressed in energetic units, (d) surface albedo $\alpha_S$, and (d) TOA upwelling SW radiation $F_{TOA}^{\uparrow}$ between 'Mid' and 'End'. Markers denote models as they are numbered in Table 1. The color scale depicts changes in asymmetry between 'End' and 'Mid'.

Models disagree on whether poleward heat transport in the SH increases or decreases between 'Mid' and 'End'; however, models that lose more cloud cover in the SH midlatitudes exhibit only increasing poleward heat transport (see Supplementary Figure S5). Separating the MHT into its components shows that increased moist AHT due to increased poleward moisture transport (Figure 4c) is the primary contributor to this strengthening poleward heat transport. Changes in tropospheric zonal and surface wind speeds (shown in Supplementary Figure S6) are indicative of a stronger poleward shift in the SH eddy-driven jet and SH midlatitude storm track in models where SH extratropical cloud cover declines more strongly after the 'Mid' period. It is also worth noting that models with greater SH extratropical cloud reductions agree on a southward shift of the ITCZ after the 'Mid' period that occurs concurrently with SH cloud reductions (Figure 2i), which is expected due to the increase in absorbed solar radiation in the SH extratropics changing the hemispheric difference in net radiative heating (Geng et al., 2022).

We find that models that lose more SH subpolar extratropical cloud cover have greater warming in the Antarctic, and in the SH overall; these models also show greater reductions in Antarctic surface and planetary albedo, and thus albedo feedbacks. However, we find poor correlation between changes in SH sea ice extent and the evolution of asymmetry between 'Mid' and

'End' (shown in Supplementary Figure S7), indicating that the processes described here primarily affect Antarctic surface warming processes (i.e. the onset and strength of SH albedo feedback) in the amount of simulation time presented here.

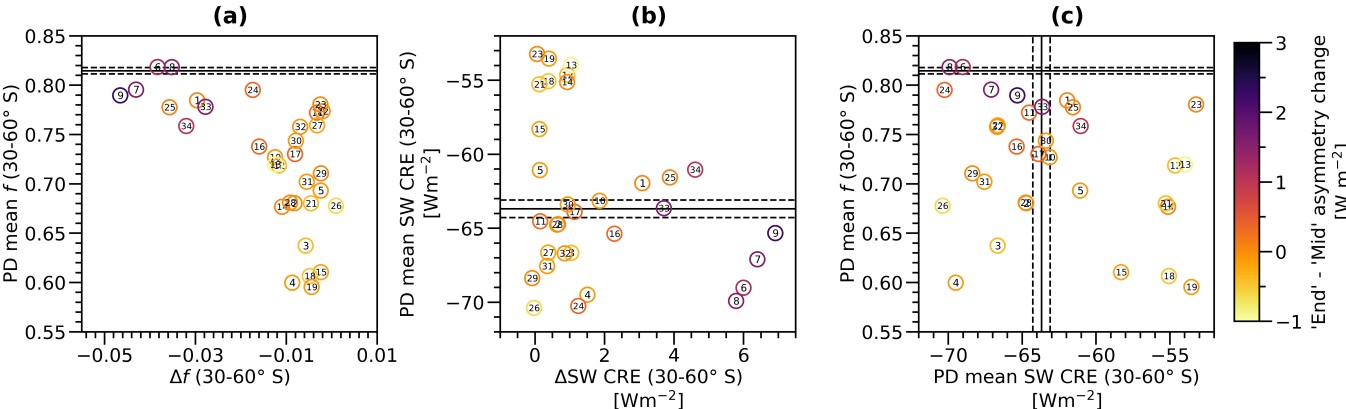

**Figure 5.** SH extratropical (30-60° S) mean (a) cloud fraction $f$ and (b) SW CRE in PD conditions plotted against differences thereof between 'End' and 'Mid'. In (c), PD mean SH extratropical $f$ is plotted against PD mean SH extratropical SW CRE. Solid black lines represent CERES EBAF (SW CRE) and MODIS (cloud fraction) mean values over March 2000-February 2015, and dashed lines represent the bounds of one standard deviation in monthly mean anomalies. The color scale depicts modeled changes in asymmetry between 'End' and 'Mid'.

Models with better representation of SH midlatitude clouds in PD conditions might also be expected to have better representations of their responses to forcing; to assess the remote compensations presented in this section, we compare model responses in models where cloud fraction and SW CRE are more similar to observations in the historical overlap period. We find that PI and PD mean cloud fraction and SW CRE are close among all models (linear regressions yield $R^2 > 0.99$ for both); therefore, we plot PD mean values of each against their responses to forcing in Figure 5. The changes in SH extratropical cloud cover outlined in this section seem to have some dependency on the model state in an unforced climate. Models with the highest cloud fraction in this region see some of the greatest reductions in cloud fraction (Figure 5a); however, the relation between mean-state SW CRE and its response to forcing is inconsistent (Figure 5b). This illustrates that it is difficult to judge whether remote compensations by SH extratropical clouds to a perturbation in hemispheric albedo asymmetry are likely or not, as a wide range of forced SW CRE responses in the SH extratropics is seen where PD mean cloud fraction and SW CRE are closest to observations (Figure 5c). Thus, these measures are not enough to estimate which response in SH extratropical clouds is more realistic.

## 3.4 The relation between SW radiative feedback strength and the albedo symmetry

In the previous three sections, our results illustrate how responses in modeled hemispheric albedo asymmetries to $CO_2$ forcing differ and diverge due to varying cloud responses in both hemispheres. Here, we demonstrate how these cloud responses are related to the strength of SW cloud radiative feedbacks.

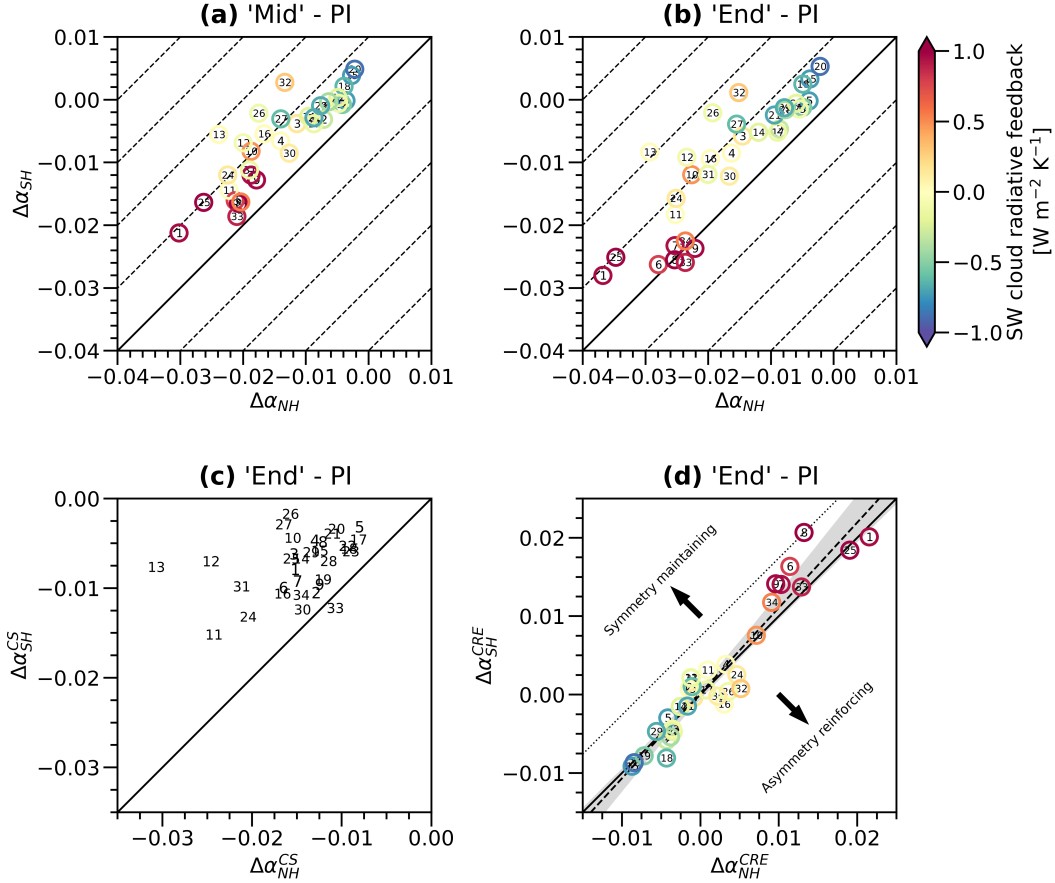

**Figure 6.** First row: changes in the NH and SH hemispheric mean planetary albedo relative to their PI mean albedo ($\Delta\alpha_{NH}$ and $\Delta\alpha_{SH}$ respectively), plotted against each other on a 'symmetry' phase space for all models in the (a) 'Mid' and (b) 'End' periods. The thick diagonal line represents symmetry scaled to PI conditions, while each dashed line parallel to this represents a 1% hemispheric difference in albedo; models on the diagonal line remain close to their PI mean asymmetry. Second row: changes in NH and SH hemispheric mean c) clear-sky albedo ($\Delta\alpha^{CS}$) and (d) SW CRE normalized by insolation ($\Delta\alpha^{CRE}$) between 'End' and PI conditions. In (d), the dotted line represents the multi-model mean clear sky albedo asymmetry anomaly by the 'End' period, and the dashed line depicts a linear regression among among the models; shading represents the 95% confidence interval for the slope. Estimates of model SW cloud radiative feedback strengths are given by the color of the marker. Markers denote models as they are numbered in Table 1.

In Figure 6a-b, we see that models that return to their pre-industrial mean asymmetry through continued SH albedo re-
ductions have stronger positive SW cloud radiative feedback strengths, associated with a stronger global mean decrease in albedo. Many models with negative or weakly positive SW cloud radiative feedback strengths remain within a 1% difference in planetary albedo after warming through lower reductions in albedo in each hemisphere, but tend to exhibit a stronger NH than SH albedo decrease. For comparison, the CERES EBAF standard deviation in the albedo asymmetry time series between 2000-2020 is 0.4 W m$^{-2}$ ($\sim$0.1%) (Jönsson and Bender, 2022), meaning that the perturbations in asymmetry due to strong

forcing in all models 150 years after the onset of abrupt $CO_2$ forcing are an order of magnitude larger than those seen in the observational record.

To investigate the degree of cloud compensations to perturbations in the albedo asymmetries, we consider what must be true in order to maintain albedo symmetry ($\Delta\delta_{HD}\alpha \approx 0$) by expressing albedo in terms consistent with equation 1:

$$\Delta\delta_{HD}\alpha = \Delta\delta_{HD}\alpha^{CS} - \Delta\delta_{HD}\alpha^{CRE} \approx 0, \qquad (2)$$

that is, the clear-sky albedo asymmetry anomaly $\Delta\delta_{HD}\alpha^{CS}$ would have to be balanced by the changes in hemispheric differences in SW CRE ($\Delta\delta_{HD}\alpha^{CRE}$), i.e.:

$$\Delta\delta_{HD}\alpha^{CRE} \approx \Delta\delta_{HD}\alpha^{CS}. \qquad (3)$$

Figure 6c shows that models agree on changes in clear-sky albedo asymmetry between 'End' and PI conditions. Here, since models are unanimous in a negative clear-sky asymmetry response to warming due to the NH darkening, we consider only the

cases where $\Delta\delta_{HD}\alpha^{CRE} > 0$ to be cases where clouds compensate for the clear-sky asymmetry anomaly. Figure 8d shows that this is usually not the case among models, and few models come close to the magnitude of $\Delta\delta_{HD}\alpha^{CRE}$ ($\sim 0.02$) that would be needed in order for clouds to fully compensate for the clear-sky albedo asymmetry. The slope of the linear fit among model SW CRE responses lies close to one, meaning that modeled SW CRE are nearly symmetrically between hemispheres; a symmetric response lies within the 95% confidence interval for the slope. That the intercept is near zero means that responses

generally act in the same direction. However, that the slope could likely be greater than one reveals a tendency for models to have a greater SW CRE response in the SH than in the NH. This slope is in the direction that would be needed in order for clouds to compensate for the perturbation in clear-sky albedo asymmetry induced by warming, but only in the case of more positive SW cloud radiative feedback.

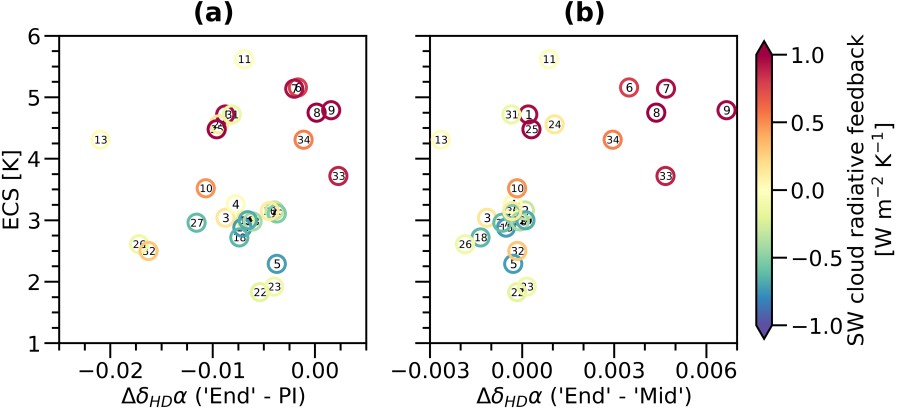

**Figure 7.** Estimates of ECS plotted against the (a) 'End' minus PI and (b) 'End' minus 'Mid' albedo asymmetry anomaly $\Delta\delta_{HD}\alpha$. Estimates of model SW cloud radiative feedback strengths are given by the color of the marker. Markers denote models as they are numbered in Table 1.

Finally, the distribution of ECS among the models in relation to their responses in albedo asymmetry to $CO_2$ forcing is shown in Figure 7. Responses in the asymmetry between PI conditions and 'End' do not reveal a relation between the total asymmetry response to warming and ECS (Figure 7a). However, the mechanism through which some models recover PI mean asymmetry – SH midlatitude cloud loss between 'Mid' and 'End' – contributes to positive SW cloud feedbacks. This seems to be consistent with those models also having amongst the highest ECS (Figure 7b), although the relation is not strong ($R^2 = 0.30$ with $p = 0.002$).

## 4 Discussion

If a lasting hemispherically asymmetric albedo response to warming is possible, intermediate ranges of total SW radiative feedback strength estimates would be possible. However, if symmetry is to be maintained in a changing climate, hemispheric differences in surface albedo changes must be compensated for by clouds in one of two ways: by local or by remote compensation. With local compensation, surface albedo reductions are partially compensated for by cloud changes, such as is the case to some degree in observed and modeled Arctic albedo responses to warming. With remote compensation, the requirement that albedo in both hemispheres declines symmetrically necessarily means that the global albedo reduction is greater. These two possibilities, local or remote compensations, would contribute negative or positive SW radiative feedbacks, respectively.

In the cases where clouds act in the direction of compensating for albedo asymmetry perturbations (Section 3.4, Figure 6), there are no cases where NH local compensations – and thus negative SW cloud feedbacks – are strong enough for models to maintain their PI albedo asymmetry. Combinations where reductions in SW CRE are larger in the SH than in the NH – and thus positive SW cloud feedbacks – do lead to instances where PI albedo asymmetries are maintained. This may mean that if clouds do compensate for disturbances in the albedo asymmetry, an effective way for this to occur is via remote compensations, and scenarios involving more local compensations are unlikely to be capable of maintaining albedo symmetry. However, because of a lack of long-term measurements of Earth's radiative energy system with which to study the evolution of Earth's hemispheric albedo symmetry throughout changes in the climate and anthropogenic forcings, detecting the strength and causal sources of these compensations is currently challenging.

The fact that models with higher SH extratropical cloud fraction — and that are thereby in better agreement with observations — lose more clouds in response to warming could be taken as indication that a greater cloud loss and hence more positive cloud feedback and remote compensation is more realistic. On the other hand, the relation between unforced or historical mean SW CRE and the change in SW CRE in response to forcing is weak, leading to an inconsistent constraint on remote compensations. Models may also have realistic cloud properties in this region for the wrong reason, and models with more cloud cover may also see greater changes simply due to having more clouds in the unforced state to lose (Kajtar et al., 2021). The link between mean state magnitude and forced response is ultimately not self-evident (McCoy et al., 2014; Zelinka et al., 2022; Kuma et al., 2022), and the time scales of the SH extratropical cloud changes involved may mean that albedo symmetry-restoring remote compensations do not act on the time span of the observations that we presently have (Frey et al., 2017; Gjermundsen et al., 2021).

A remaining question is whether SH extratropical cloud reductions are to be seen in more models beyond the 150 years that are the required minimum simulation time for CMIP6 strong forcing experiments. Gjermundsen et al. (2021) found that despite NorESM2 having weaker SW cloud radiative feedbacks than CESM2 at year 150, the strength of these feedbacks are comparable after 500 years of simulation when SO SSTs have risen to a similar degree. Previous results based on a single model show that SO deep convection also influences the occurrence of Antarctic warm events in a climate at equilibrium through processes similar to those presented in Section 3.3 (Pedro et al., 2016). It is possible that more models follow this pattern – and thereby exhibit remote compensations – on longer timescales. The sensitivity of modeled clouds to SSTs in this region would point to changes in SO SSTs being able to induce a SH albedo reduction and thus remote compensations. The overturning circulation that impacts SSTs in this region reaches between hemispheres and could thus present a cross-hemispheric communicator for energetic anomalies caused by changes in albedo.

Here we have also shown that in models, if a remote compensation to NH albedo reductions is accomplished by SH albedo reductions, the ITCZ will consequently move southward, in line with studies on changes in tropical precipitation under warming (e.g. Geng et al. 2022). If reductions in SH extratropical cloud cover and Antarctic albedo reinforce hemispheric differences in net radiative heating, a southward shift of the ITCZ may also occur and thus negate some of the remote compensation to asymmetry anomalies offered by SH extratropical albedo reductions, which may require more substantial extratropical albedo reductions to maintain hemispheric albedo symmetry. Although tropical clouds and albedo seem to play a secondary role in determining the observed hemispheric albedo symmetry on time scales longer than a year (Bender et al., 2017; Datseris and Stevens, 2021; Jönsson and Bender, 2022), this should also be taken into account in understanding hemispheric albedo symmetry-maintaining mechanisms that involve the extratropics. However, model representations of tropical clouds, bias in the ITCZ position, and spurious, unrealistic ITCZ dynamics complicate projections of tropical clouds and albedo in a changing climate (e.g. Hwang and Frierson 2013; Zhang et al. 2019a; Tian and Dong 2020).

## 5   Conclusions

Following the observation that Earth's albedo is persistently hemispherically symmetric throughout the most recent two decades of satellite observations, we investigated responses in hemispheric differences in albedo to $CO_2$ forcing, and their implications for cloud feedback, heat redistribution, and spatial patterns of warming. To do this, we made use of the evolution of hemispheric albedo differences in CMIP6 models when $CO_2$ concentrations are abruptly quadrupled.

In all models, NH albedo is immediately reduced due to albedo reductions in mid- to high latitudes following ice loss and cloud changes, causing the hemispheric difference in albedo to be SH-favored relative to PI conditions consistently among all models. However, models do not agree on the strength of this initial response. In some models, cloud cover increases in the NH to reduce the impact of clear-sky albedo reductions in the hemispheric mean, and in others, changes in NH cloud cover strengthens the all-sky albedo reductions. The former represents one way in which hemispheric albedo symmetry may be maintained: local compensations to albedo reductions.

Another possibility for maintaining hemispheric albedo symmetry involves remote compensations to hemispherically asymmetric albedo reductions: in some models, asymmetry states close to their PI conditions are restored when SH albedo is reduced, primarily via reductions in SH extratropical cloud cover. Here, we have shown that changes in SH extratropical cloud cover are linked with Antarctic polar amplification and changes in Antarctic albedo. When cloud cover is reduced in the SH extratropics, the increased absorbed energy is redistributed polewards, contributing to the spread in modeled Antarctic responses to $CO_2$ forcing.

These two pathways illustrate how mechanisms maintaining the hemispheric albedo symmetry impact the climate sensitivity through their implications for SW radiative feedbacks. Depending on the degree of local compensation (increasing total cloud contributions to albedo) and remote compensation (decreasing total cloud contributions to albedo), the implied SW cloud radiative feedback can be either negative or positive. Our results show that clouds may serve to suppress an asymmetric response in the hemispheric albedo difference to forcing so that the all-sky albedo is more hemispherically symmetric than clear-sky albedo, but may not necessarily fully compensate for a perturbed hemispheric albedo difference. The lack of observational constraints or evidence for the existence of a symmetry maintaining mechanism limits the possibility to evaluate model realism based on their degree of symmetry restoration. The lack of consistent pathways for symmetry restoration among the models limits the possibility to assess a single mechanism for keeping the symmetry. However, there is clearly an implication for the strength of cloud-climate feedbacks depending on whether a possible mechanism that maintains the hemispheric albedo symmetry involves cross-hemispheric communication or not.

*Data availability.*  The CMIP6 model output used for this study can be accessed through the Earth System Grid Federation (ESGF).

*Author contributions.*  A.R.J.: conceptualization, formal analysis, investigation, methodology, software, visualization, and writing (original draft preparation, reviewing, and editing). F.A.-M.B.: conceptualization, funding acquisition, methodology, project administration, supervision, and writing (review and editing).

*Competing interests.*  The authors declare no competing interests.

*Acknowledgements.*  This research is part of a project funded by the Swedish Research Council (Grant 2018-04274). We thank the World Climate Research Programme and its Working Group on Coupled Modelling for coordinating and promoting CMIP. We also acknowledge and thank the German Climate Computing Centre (DKRZ) and the European Union's Horizon 2020 "Infrastructure for the European Network for Earth System Modelling phase 3" (IS-ENES3) project (grant agreement No. 824084) for providing access to computational resources and CMIP6 model output. Finally, for their helpful advice in discussions and comments on earlier versions of this manuscript, we would like to thank Thorsten Mauritsen, Maria Rugenstein, George Datseris, Michael Diamond, and Aiko Voigt.

| | Model | Citation (abrupt-4xCO2, piControl, historical) |
|---|---|---|
| 1) | ACCESS-CM2 | Dix et al. (2019a, b, c) |
| 2) | AWI-CM-1-1-MR | Semmler et al. (2018a, c, b) |
| 3) | BCC-CSM2-MR | Wu et al. (2018a, c, b) |
| 4) | BCC-ESM1 | Zhang et al. (2019b, 2018b, a) |
| 5) | CAMS-CSM1-0 | Rong (2019a, c, b) |
| 6) | CESM2 | Danabasoglu (2019a); Danabasoglu et al. (2019); Danabasoglu (2019d) |
| 7) | CESM2-FV2 | Danabasoglu (2020a, 2019g, c) |
| 8) | CESM2-WACCM | Danabasoglu (2019b, h, f) |
| 9) | CESM2-WACCM-FV2 | Danabasoglu (2020b, 2019i, e) |
| 10) | CMCC-CM2-SR5 | Lovato and Peano (2020a, c, b) |
| 11) | CanESM5 | Swart et al. (2019a, c, b) |
| 12) | EC-Earth3-AerChem | EC-Earth Consortium (2020a, c, b) |
| 13) | EC-Earth3-Veg | EC-Earth Consortium (2019a, c, b) |
| 14) | FGOALS-f3-L | Yu (2019a, c, b) |
| 15) | FGOALS-g3 | Li (2019a, c, b) |
| 16) | GFDL-CM4 | Guo et al. (2018a, b, c) |
| 17) | GFDL-ESM4 | Krasting et al. (2018a, b, c) |
| 18) | GISS-E2-1-G | NASA Goddard Institute for Space Studies (2018a, c, b) |
| 19) | GISS-E2-1-H | NASA Goddard Institute for Space Studies (2019a, 2018d, 2019b) |
| 20) | GISS-E2-2-G | NASA Goddard Institute for Space Studies (2019c, d) |
| 21) | IITM-ESM | Gopinathan et al. (2019); Narayanasetti et al. (2019); Choudhury et al. (2019) |
| 22) | INM-CM4-8 | Volodin et al. (2019a, c, b) |
| 23) | INM-CM5-0 | Volodin et al. (2019d, f, e) |
| 24) | IPSL-CM6A-LR | Boucher et al. (2018a, c, b) |
| 25) | KACE-1-0-G | Byun et al. (2019a, c, b) |
| 26) | MIROC6 | Tatebe and Watanabe (2018a, c, b) |
| 27) | MPI-ESM-1-2-HAM | Neubauer et al. (2019b, c, a) |
| 28) | MPI-ESM1-2-HR | Jungclaus et al. (2019a, b, c) |
| 29) | MPI-ESM1-2-LR | Wieners et al. (2019a, b, c) |
| 30) | MRI-ESM2-0 | Yukimoto et al. (2019b, c, a) |
| 31) | NESM3 | Cao and Wang (2019a, c, b) |
| 32) | NorESM2-MM | Bentsen et al. (2019a, c, b) |
| 33) | SAM0-UNICON | Park and Shin (2019a, c, b) |
| 34) | TaiESM1 | Lee and Liang (2020a, c, b) |

**Table 1.** CMIP6 member models used in this study, and their representative numbers when model number is displayed in figures.

# Appendix A:  Model variables used in this study

| Variable | CMIP6 output variable name |
| --- | :---: |
| Upwelling SW radiative flux at TOA (all-sky) | rsut |
| Upwelling SW radiative flux at TOA (clear-sky) | rsutcs |
| Incoming SW radiative flux at TOA | rsdt |
| Outgoing LW radiative flux at TOA | rlut |
| Net downward radiative flux at TOA | rtmt |
| Surface upwelling SW radiative flux | rsus |
| Surface downwelling SW radiative flux | rsds |
| Surface upwelling LW radiative flux | rlus |
| Surface downwelling LW radiative flux | rlds |
| Surface upward sensible heat flux | hfss |
| Surface upward latent heat flux | hfls |
| Cloud area fraction | clt |
| Vertically integrated atmospheric cloud condensed water content | clwvi |
| Vertically integrated cloud ice content | clivi |
| Vertically integrated atmospheric water vapor content | prw |
| Total precipitation | pr |
| Ice-phase precipitation | prsn |
| Total evaporatranspiration and sublimation | evspsbl |
| Near-surface (10 m) wind speed | sfcWind |
| Eastward wind speed | ua |
| Sea ice area concentration | siconc |

**Table A1.** Variables used in this study and their CMIP6 standard short names.

| | Model | Variables included | | | | | | |
|---|---|---|---|---|---|---|---|---|
| | | clt | clwvi, clivi | prw | prsn | evspsbl | sfcWind | siconc |
| 1) | ACCESS-CM2 | + | | + | + | + | + | + |
| 2) | AWI-CM-1-1-MR | + | + | + | + | + | + | |
| 3) | BCC-CSM2-MR | + | + | + | + | + | + | |
| 4) | BCC-ESM1 | + | + | + | + | + | + | |
| 5) | CAMS-CSM1-0 | + | + | + | | + | + | + |
| 6) | CESM2 | + | + | + | | + | + | + |
| 7) | CESM2-FV2 | + | | + | + | + | + | + |
| 8) | CESM2-WACCM | + | + | + | | + | + | + |
| 9) | CESM2-WACCM-FV2 | + | | + | + | + | + | + |
| 10) | CMCC-CM2-SR5 | + | + | | + | + | + | + |
| 11) | CanESM5 | + | + | + | + | + | + | + |
| 12) | EC-Earth3-AerChem | + | | + | + | + | | + |
| 13) | EC-Earth3-Veg | + | | + | + | | | |
| 14) | FGOALS-f3-L | + | + | | + | + | + | |
| 15) | FGOALS-g3 | + | | + | + | + | + | |
| 16) | GFDL-CM4 | + | + | + | + | + | + | + |
| 17) | GFDL-ESM4 | + | + | + | + | + | + | + |

**Table A2.** Model output coverage in this study. Only variables where output was missing from some models are listed, and all other variables listed in Table A1 that are not present here are fully included in the study. Models where variable output was available for all experiments and presented in the study are marked with a plus sign (+).

| | Model | Variables included | | | | | | |
|---|---|---|---|---|---|---|---|---|
| | | clt | clwvi, clivi | prw | prsn | evspsbl | sfcWind | siconc |
| 18) | GISS-E2-1-G | + | + | + | + | + | + | |
| 19) | GISS-E2-1-H | + | + | + | + | + | + | + |
| 20) | GISS-E2-2-G | - | | + | + | + | + | |
| 21) | IITM-ESM | + | + | | + | + | + | + |
| 22) | INM-CM4-8 | + | + | + | + | + | + | + |
| 23) | INM-CM5-0 | + | + | + | + | + | + | + |
| 24) | IPSL-CM6A-LR | + | + | + | + | + | + | + |
| 25) | KACE-1-0-G | + | + | + | + | + | + | |
| 26) | MIROC6 | + | + | + | + | + | + | + |
| 27) | MPI-ESM-1-2-HAM | + | + | + | + | + | + | + |
| 28) | MPI-ESM1-2-HR | + | + | + | + | + | + | + |
| 29) | MPI-ESM1-2-LR | + | + | + | + | + | + | + |
| 30) | MRI-ESM2-0 | + | + | + | + | + | + | + |
| 31) | NESM3 | + | + | + | + | + | | + |
| 32) | NorESM2-MM | + | + | + | + | + | + | + |
| 33) | SAM0-UNICON | + | + | + | + | + | + | + |
| 34) | TaiESM1 | + | + | + | + | + | + | |

**Table A3.** Model output coverage in this study, continued from Table A2. Only variables where output was missing from some models are listed, and all other variables listed in Table A1 that are not present here are fully included in the study. Models where variable output was available for all experiments and presented in the study are marked with a plus sign (+), and a dash (-) is used where output from historical simulations was missing (only one variable, clt, from one model, GISS-E2-2-G).

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
