# Peer review of "The implications of maintaining Earth's hemispheric albedo symmetry for shortwave radiative feedbacks"

_EGUsphere, 2022_

## Author Comment (AC1)

For the submission: "The response of hemispheric differences in Earth's albedo to CO2 forcing in coupled models and its implications for shortwave radiative feedback strength" (https://doi.org/10.5194/egusphere-2022-811)

Aiden Jönsson, Frida Bender

Below: responses by the authors are in standard text, and the quoted reviewer comments are italicized.

**Reviewer 1**

*This manuscript frames itself as exploring the possibilities of two future scenarios: one where the hemispheric albedo symmetry persists, and one where it is does not. Unfortunately, there is no clear delineation between these two regimes in the model results. Looking at the model spread in hemispheric albedo difference changes, it appears to be a normal distribution centered around -3 W/m2 (Figure 1). There are a few cases overlapping 0 W/m2, but these may simply be due to chance. All models show an initial negative perturbation to hemispheric albedo difference owing to a reduction in clear-sky albedo in the NH high latitudes. For some models, this is partially offset by increases in NH cloud albedo (termed a local compensation). For other models, the NH albedo reduction is matched by a reduction in SH albedo – mostly by clouds in the latitude range 30-60S – termed a remote compensation. For the remaining models, there is little compensation by clouds and the hemispheric albedo difference simply persists. It does not appear that there is any strong correlation between the magnitude of hemispheric symmetry change and SW cloud feedback (as shown in Figure 8), making it difficult to conclude anything about the role hemispheric albedo difference plays for climate projection uncertainty.*

*It is hard to gauge what we, as readers, are to learn from this study. It seems the conclusions are limited beyond acknowledging there is a large spread in model behavior surrounding clouds, which is already well known. If anything, this work would seem to suggest that models have no inclination for maintaining a hemispheric albedo symmetry, which makes sense given that they do not generally reproduce the observed symmetry (as shown in Supplementary Figure S1). The authors acknowledge that no physical mechanism has been proposed for why the hemispheric albedos should remain balanced, so it is perhaps not surprising that the models are unconstrained for their own hemispheric albedo differences. Some of the relationships examined in this manuscript between the cloud changes and other physical processes in the models may be useful to the scientific community, but I found the interpretation questionable at times (I have added details to these issues below). In its present form, I must recommend this paper be rejected and returned to the authors.*

We would first like to thank the reviewer for their time and effort in reading and commenting on our manuscript. The reviewer points out issues with the manuscript in its present form, and we wish to address these concerns in a revised draft. The concrete suggestions as well as the reviewer's descriptions of their reading experience are much appreciated.

Our goal is not to separate models into groups where albedo symmetry is and is not maintained, or to compare two competing scenarios, or to say which scenario is more realistic. As the reviewer stated, there is no clear differentiation between models in this

regard, nor is there currently an appropriate way to constrain which response is most realistic. However, we find that the models behave differently in terms of the response of albedo asymmetry to warming, creating a spectrum of possible responses. After the initial response, on which models agree relatively well, models could be grouped into categories of "strengthening asymmetry", "retaining a new asymmetry", and "restoring pre-industrial asymmetry". Because thresholds for any such groups will be subjective and to some extent arbitrary, we rather present the responses as a spectrum. The fact that these coupled models can be pushed out of their initial state of asymmetry, and that some to varying degrees return, is an interesting point in itself that to our knowledge has not been discussed in the literature.

With our analysis, we illustrate how restoration can have different implications for cloud radiative feedbacks, and we can also see that the primary feature that the asymmetry-restoring models have in common is a SH cloud loss that implies positive cloud feedback. The local NH increase in cloud covers that contribute to restoration – and to negative cloud feedbacks – are not sufficient to restore the original state of symmetry.

Since the question of whether there is a restoring mechanism in reality – and if so, what its drivers are – is still unanswered, we want to use the models' responses to strong forcing to illustrate potential pathways to albedo symmetry restoration, and quantify their impact on feedback. There are to date no fitting observational constraints for the model behavior, but as the current observational record extends in time, there can be an indication of which of the behaviors is more consistent with the actual climate's response, making it useful to have mapped those pathways and their impact on cloud feedback beforehand.

We appreciate the comment that the manuscript is too lengthy, and with that lacking in focus. Therefore, we have made significant changes in the scope of the main text (by moving some of the results to the supplementary material) and sharpened the focus on the key message by restructuring. We have also changed the title to be more concise, and to more effectively convey what the focus of our manuscript is. We would once again like to thank the reviewer for their guidance and input in focusing our text, and invite them to read our revised manuscript.

**Major issues**

*Eyeballing the 'end' period in Fig 1a, it seems like the models suggest a normal distribution of asymmetry changes centered around -3 W/m2. Those models that come in at 0 W/m2 change seem to do so by chance. I attempted to go through the various tables and figures to determine if the models that start with a symmetric albedo (Figure S1) are the same ones that have a small 'End' – 'PI' hemispheric albedo difference, but I couldn't find such a relationship. Do the models that have a small perturbation change to warming at the end of 150 years have any consistent relation to their initial hemispheric albedo difference? Is there any change in the distribution across models of hemispheric differences with warming?*

While there is a spread in initial degrees of albedo asymmetry, we find no significant relation between the response and the initial asymmetry. We thank the reviewer for posing this question, and have explicitly stated this in the text (lines 139-141), as well as included the previous Supplementary Figure S1 in the revised manuscript's Figure 1 with the mean asymmetries of several periods in order to help illustrate this.

*Line 128: "While models agree on clear-sky albedo reductions in the NH in response to warming, the spread in magnitude of total albedo reductions points to differences among the models in whether clouds serve to either amplify or reduce the total albedo reduction in the hemispheric mean." There is very little agreement in the magnitude of clear-sky albedo change in response to warming (Fig 2b). Are the authors arguing that clouds determine how much sea ice is lost? How do we know that is the case? Comparing Figs 2a and 2b, it appears that the spread in total albedo change at 90N is smaller than the clear-sky change. Wouldn't such a result suggest that the clouds are generally offsetting the clear-sky response to minimize the change (like the local compensation discussed later)?*

While clouds offset the clear-sky response in the Arctic, models do not agree on whether clouds elsewhere in the NH amplify the asymmetry response by further reducing NH mean albedo. This spread occurs outside of the Arctic; for this reason, we have chosen to present NH cloud changes differently in Section 3.2 and in Figure 3 by dividing the NH into three ranges of latitudes. We thank the reviewer for raising this question and prompting us to reformulate and describe NH cloud responses in a more detailed way.

*Lines 156-157: 17 models amplify and 16 reduce. There are 34 models… so 1 has no significant response? Looking at Fig 3a, it appears several of these bars are almost unreadably small. Is it really only one model where SW CRE change is not statistically distinguishable from zero?*

The effect of NH cloud responses on albedo ranges from reducing to amplifying the albedo reductions, with several having virtually no effect, although we did not test whether the distribution across models is statistically different from zero. However, we find that the previous method of visualizing and presenting NH changes in SW CRE was confusing and not informative, and have opted to present cloud responses in the NH differently in Section 3.2 of the revised manuscript.

*Line 209: "Planetary albedo is reduced in the Antarctic sea ice zone (Figure 6a); this is most likely the result of increasing liquid-phase precipitation reducing the sea ice surface albedo, and decreasing snowfall that otherwise would stabilize sea ice albedo." Why not simply a result of changing temperature or ocean circulation? I struggle to understand from the results shown how we can conclude the phase of precipitation falling on sea ice is the "most likely" cause of the albedo changes there. I see that SSTs are brought up in section 4, but I think it would be valuable to bring these changes into the discussion in section 3.3.*

We agree with and thank the reviewer for problematizing this explanation; it is true that the root of these responses is warming; to this end, we have rewritten Section 3.3, focusing on associations rather than cause-and-effect processes. We also agree that it would be helpful to place the SH midlatitude cloud reductions in the context of SSTs, which we have done by referring to the findings of Gjermundsen et al. (2021) (lines 176-180).

*Line 211: "This allows the sea ice albedo feedback to affect the SH polar climate in models where SH extratropical SW CRE increases more strongly; the result can be seen in increased SW radiative heating at the surface (Figure 6b, e)." How do we know causality here? I don't follow how Figure 6 demonstrates the SH polar changes' impact on the extratropical response.*

By necessity, a surface albedo reduction under warming indicates that a surface albedo feedback is in effect, and we find that this effect is more active in models where SH extratropical cloud reductions are greater. We believe that this is due to greater warming in the SH midlatitudes leading to increased poleward heat transport, impacting Antarctic climate. We do not claim that SH polar changes are impacting the extratropical response, but rather the other way around, and have ensured that the chain of reasoning is consistent in the revised manuscript to avoid this confusion. We thank the reviewer for pointing out where this confusion lies, and have rewritten Section 3.3 in order to convey this finding more effectively.

*Section 3.3: I struggle to follow the argument of the poor correlation between sea ice extent and changes in extratropical SW CRE changes. Why are the authors only looking at changes in maximum sea ice extent? Why not some time-integrated sea ice extent measure? Wouldn't the sea ice minimum be more interesting because a larger retreat during summer would have impacts on surface fluxes that could change the clouds and circulation patterns nearby? All the changes in clouds have been annual averages, so why compare them with a seasonally dependent measure of sea-ice?*

We agree that the measure of annual maximum sea ice extent is a limited measure of annual mean albedo, and find the reviewer's suggestion to quantify time-integrated sea ice extent to be helpful, as it could capture the potential effect on Antarctic sea ice throughout the year. This has been added to Supplementary Figure S7, and we found that the annually integrated sea ice extent also does not have a strong control on the 'Mid'-'End' asymmetry evolution. We thank the reviewer for this suggestion.

*Line 244: "…meaning that the perturbation in asymmetry due to strong forcing in all models 150 years after the onset of abrupt $CO_2$ forcing is close to the interannual variability seen in the past 20 years of observations." If all models are close to the interannual variability, what does that tell us? How do we reconcile that result with the discussion around Figure 1?*

The original sentence referred to here was incorrect, and the interannual variability is indeed less than was reported in the original manuscript. The perturbations are an order of magnitude larger than that that is seen in interannual variability in the observational record, and are therefore significant in comparison. We thank the reviewer for pointing out this discrepancy, and have changed the text to reflect the correct comparison in magnitudes (lines 222-225).

*Line 255: "When the difference between NH and SH $\Delta(\alpha clear - \alpha)$ is larger, asymmetry is more effectively maintained." Is this true? Eyeball estimates in Figure 8c don't show a clear signal. Is this plotted somewhere else or has a correlation been computed?*

We agree that this presentation was unclear, and have opted to use a different method and visualization to explain the impact of hemispheric differences in SW CRE on the asymmetry response. We believe that the current explanation will more easily convey that

the SW CRE response must itself be hemispherically asymmetric in order to maintain PI asymmetry (Figure 6, lines 238-242).

**Minor issues**

*Line 163 "on both the the degree" -> "on both the degree"*

This sentence has been reformulated in the revised manuscript (lines 161-165).

*Figure 4a – is the colorscale reversed here? How do the lines peaking over +10 W/m2 have an average of -1 W/m2? The caption text suggests they are the same variable differencing the same time periods. It doesn't match Figure 5 either.*

The color scale was indeed reversed in this figure, and read therefore incorrectly. We are thankful for the reviewer catching this; however, we have removed this plot from the revised manuscript due to redundancy (as it has already been shown in Figure 2).

*Figure 4b-f are the bounds too narrow on these plots? Where are models 9, 7, and 1 in panels c-f?*

Output for cloud water content for models 9, 7, and 1 were missing or corrupted and thus not included in the scatter plots for cloud water content, which is shown in Tables A2 and A3. We have now emphasized that certain variables are not covered in the Methods and materials section (Section 2.1; lines 89-91).

*"We henceforth use the difference in 30-60° S area mean SW CRE between the 'End' and 'Mid' periods as an indicator of the impact of cloud albedo contribution changes on TOA albedo in the SH extratropics among models." Is 30-60S SW CRE well-correlated with the total SH SW CRE change? In other words, is it fair to focus on this region because variability here corresponds to the total variability we are concerned with (the remote/SH albedo changes)?*

The difference in 30-60° S area mean SW CRE is strongly related to the evolution of asymmetry between 'Mid' and 'End', but the reviewer is right in pointing out that they do not encompass all changes. Therefore, we have changed the indicator to instead refer to the difference in asymmetry between 'End' and 'Mid' referred to in Section 3.1 in order to be internally consistent, and thank the reviewer for bringing this up.

*"Note also that SW CRE at higher latitudes (> 60° S) also becomes more negative consistently in models with SW CRE increases in the SH extratropics." Is poleward of 60S considered extratropics here?*

We agree that the wording should reflect that the polar region is excluded from the observation here, and we have revised the manuscript to specify whenever the poles are not meant to be referred to (e.g. subpolar extratropical and midlatitude).

*"net poleward transport of moisture away from the SH extratropics (~30-50° S) to the polar region (> 60° S)" Now extratropics is 30-50S?*

This was a typo, and was meant to read 30-60° S; however, this sentence has been removed in the revised manuscript.

*"Atmospheric moisture content increases in the SH (Figure 5a) as clouds are lost and the atmosphere is warmed." -> This reads as if the cloud loss helps cause the increase in atmospheric moisture, which I am guessing the authors did not mean to imply.*

The reviewer is correct, and we thank them for catching this; we have specified in the revised manuscript that the increase in moisture is due to warming (lines 183-185), but occurs more strongly in models with greater SH midlatitude cloud loss.

*Figure 7 – colorbar is flipped again?*

The colorbar direction has been corrected in the revised manuscript and is consistent with the indicator of 'End'-'Mid' asymmetry changes as stated earlier; we thank the reviewer for catching this.

*Line 268: "These two possibilities, local or remote compensation, would also mean that SW radiative feedback strengths are either strongly positive or somewhat negative, respectively." Isn't this flipped? Remote compensation has the strong positive SW radiative feedback.*

We thank the reviewer for catching this mistake, and have revised the manuscript to correctly refer to the local and remote compensations as negative and positive feedbacks, respectively (lines 255-256).

*Line 290: "role in determining the the observed" -> "role in determining the observed"*

This typo has been corrected in the revised manuscript (lines 291-292).

*"Although tropical clouds and albedo seem to play a secondary role in determining the observed hemispheric albedo symmetry on time scales longer than a year, this should also be taken into account in understanding hemispheric albedo symmetry-maintaining mechanisms that involve the extratropics, as it can mean that some of the compensation offered by extratropical albedo reductions in one hemisphere can be buffered by tropical albedo increases, which may require more substantial high latitude albedo reductions to maintain hemispheric albedo symmetry." -> should be separated into multiple sentences for clarity*

We agree with the need to separate this sentence for clarity, and thank the reviewer for the suggestion; this has been done in the revised manuscript (lines 291-296).

*Appendix B and Figures B1-B3 are not referenced anywhere in the text.*

We have opted to move Figures B1-B3 to the Supplementary Material.

---

## Author Comment (AC2)

For the submission: "The response of hemispheric differences in Earth's albedo to CO2 forcing in coupled models and its implications for shortwave radiative feedback strength" (https://doi.org/10.5194/egusphere-2022-811)

Aiden Jönsson, Frida Bender

Below: responses by the authors are in standard text, and the quoted reviewer comments are italicized.

**Reviewer 2**

*Jönsson and Bender explore changes in albedo, radiative fluxes and cloudiness in order to improve the understanding of the hemispheric symmetry of the planetary albedo and its possible changes in a warming climate. This is performed by investigating output from the CMIP6 in combination at some points with satellite retrievals. The topic of hemispheric symmetry of the planetary albedo is an exciting and highly debated one, in particular in light of possible changes in a warming climate. The study is of interest to the readership of EGUSphere. It is written in excellent English and the figures are in good quality.*

*The analysis is thoroughly conducted and broad in scope. In fact, my most important remark is that there is so much material that at multiple times I was a bit lost in understanding as to how a particular result allows to conclude about the causes for changes in hemispheric difference in planetary albedo.*

*The Discussion section is excellent, but does not really discuss the results in light of the literature. It would rather be better as part of the Introduction, and then the discussion of the results could refer to it. I do not provide a specific suggestion for shortening the results sections, but I propose the authors consider moving some of the material to an annex to streamline the discussion.*

We would like to thank the reviewer for their critical reading and comments on our manuscript. We especially appreciate the reviewer's suggestion in focusing the manuscript in order to communicate our main points more effectively. With that, we recognize the need to guide the reader in a better way. Specifically, we think that the suggestions to move many of the details of the analysis to the supplementary material as well as to introduce some of the literature material in the Discussion section in the Introduction section are very helpful, and for this we are grateful. Beyond this, we have also changed the title of the manuscript in order to more effectively and concisely convey its focus.

We feel that the reviewer's opinions shared on our manuscript have greatly helped to improve it. With these suggestions in mind, we have revised our manuscript and would like to invite the reviewer to read.

*Besides this, I only have a number of specific remarks.*

*l72 It is regressing the global mean temperature against the top-of-atmosphere radiation imbalance (the effect forcing is the y-axis intercept only)*

This has been corrected to refer to the net TOA radiation imbalance as the regression variable (lines 77-79) in the revised manuscript; we thank the reviewer for catching this.

*l88 I find this definition throughout the manuscript puzzling, since now all signs for CRE are the opposite ones compared to the all-sky and clear-sky differences. I think this definition requires that in Fig. 2, Fig 3 etc the reader is reminded about this difference in definition.*

We have used the CRE convention of clear-sky minus all-sky fluxes, which gives SW CRE to be negative (in order to be associated with reduced absorption of SW radiation and thus cooling). We do understand the potential for confusion and agree that the reader should be reminded of how changes in SW CRE correspond to changes in absorption/reflection, and have adjusted the figure captions to remind the reader of this definition, as well as its introduction in the Section 3.2. We thank the reviewer for suggesting to remind the reader of the direction of effects.

*l154 (Fig. 1 and subsequent similar figures) – it would be useful to colour the numbers in the scatterplots by the colour used for the corresponding lines in the line plot to allow to make the association at least vaguely.*

We agree that using consistent color scales throughout the text would make the association easier to make, and have adjusted color scales throughout the manuscript to be consistent with Section 3.1.

*l159 I propose it might be better to use the same y-axis scaling in all panels*

We agree with the reviewer and thank them for this suggestion; this has been implemented in the revised manuscript's Figure 2.

*l173 I do not understand Fig. 3b. The three bars for each model should add up. Why is that not the case? also: Clarify in Caption that this is the difference between mid and PI*

The reviewer is correct that the bars should add up, and thank them for catching this mistake; the original figure had errors in the data processing, and we did not catch this. However, this figure has been changed to a different form in the revised manuscript in order to make the analysis easier to comprehend. That the difference is between 'Mid' and PI has been clarified in the new Figure 3's caption, and we thank the reviewer for this suggestion.

*l202 Would it maybe be interesting to express precipitation and e-p in energetic units for comparison to the SW fluxes? Are the authors sure about no mistake for the models that substantially cool the NH high latitudes between mid and end?*

We agree that energetic units would make more sense for this comparison, since the argument is based on heat transport considerations, and have adjusted Figure 4 accordingly. While no mistake was found in the calculations for the previous figure, since we focus only on SH processes in Section 3.3, we have opted to quantify only area averages for the relevant SH regions in Figure 4 in the revised manuscript rather than zonal mean profiles.

*l209 What is cause and what is effect is not fully clear. It may also be that after sea ice melting, clouds are much warmer if connected to the warm ocean rather than cold sea ice. Maybe reformulate to "this is most likely related to"*

It is true that cause and effect are difficult to distinguish here, and have rewritten Section 3.3 in order to emphasize that the changes in SH climate variables and heat transports that we find are associations. While the sentence that this comment refers to has been removed in the revised manuscript, we have taken the wording into account in the rewrite, and thank the reviewer for this comment.

*l212 To me it is not clear enough why Fig. 6b,e are not largely redundant with Fig. 6a,d*

We agree that Figure 6's profiles are largely redundant, and have opted to remove Figure 6 and combine its most relevant components (scatter plots showing the change in 30-60° S mean SW CRE change against > 60° S Antarctic changes) into Figure 4. We are thankful to the reviewer for pointing this out.

*l215 What exactly are the "conditions" if not extent of sea ice?*

For a given sea ice extent, surface albedo may vary and thus indicate the presence of albedo feedbacks in effect, which we find to be occurring more strongly in models with greater SH extratropical cloud reductions. However, we agree that sea ice is the primary condition to investigate when considering Antarctic albedo and that further checks would be useful in addressing this, which Reviewer 1 pointed out and suggested; to this end, we have also plotted asymmetry changes against annually integrated sea ice extent and sea ice extent maxima, which we found to not have a strong control on the differences in asymmetry between 'Mid' and 'End' (see Supplementary Figure S7). We thank the reviewer for posing this question.

*l222 I would formulate the other way around, y-axis plotted against x-axis. Clarify that cloud fraction is from MODIS, not CERES.*

The suggestion to plot the present-day values on the y-axis has been taken into account in the revised Figure 5, and cloud fraction has been attributed properly to MODIS in the revised manuscript (in the Methods and materials section, 2.1, and in the caption of Figure 5).

*l245 "within" rather than "close to", I guess, since many models have lower values.*

This sentence contained a mistake in comparing the interannual variability within the observed albedo symmetry time series; this has been corrected in the revised manuscript. The responses are an order of magnitude larger than perturbations on the interannual timescale, and are therefore significant (lines 222-225). We are thankful to the reviewer for pointing this out.

*l315 This "model dependence" I do not understand. Of course the models show different results, so the results are model-dependent. What exactly is meant, a specific influence of the dynamical core of CESM?*

We agree that this formulation is unclear and that the statement of model dependence is trivial, and have opted to remove this sentence; we thank the reviewer for commenting on this.